# Morphogenesis of extra-embryonic tissues directs the remodelling of the mouse embryo at implantation

Neophytos Christodoulou [1,6], Antonia Weberling[1,6], Douglas Strathdee [2], Kurt I. Anderson [3], Paul Timpson[4,5] & Magdalena Zernicka-Goetz [1]

Mammalian embryos change shape dramatically upon implantation. The cellular and molecular mechanism underlying this transition are largely unknown. Here, we show that this transition is directed by cross talk between the embryonic epiblast and the first extra-embryonic tissue, the trophectoderm. Specifically, we show via visualisation of a Cdx2-GFP reporter line and pharmacologically mediated loss and gain of function experiments that the epiblast provides FGF signal that results in differential fate acquisition in the multipotent trophectoderm leading to the formation of a tissue boundary within this tissue. The trophectoderm boundary becomes essential for expansion of the tissue into a multi-layered epithelium. Folding of this multi-layered trophectoderm induces spreading of the second extra-embryonic tissue, the primitive endoderm. Together, these events remodel the pre-implantation embryo into its post-implantation cylindrical shape. Our findings uncover how communication between embryonic and extra-embryonic tissues provides positional cues to drive shape changes in mammalian development during implantation.

[1] Mammalian Embryo and Stem Cell Group, University of Cambridge, Department of Physiology, Development and Neuroscience, Downing Street, Cambridge CB2 3DY, UK. [2] Cancer Research UK Beatson Institute, Glasgow G61 1BD, UK. [3] Francis Crick Institute, London NW1 1AT, UK. [4] Garvan Institute of Medical Research and The Kinghorn Cancer Centre, Cancer Division, Sydney, NSW 2010, Australia. [5] St Vincent's Clinical School, Faculty of Medicine, University of NSW, Sydney, NSW 2010, Australia. [6]These authors contributed equally: Neophytos Christodoulou, Antonia Weberling. Correspondence and requests for materials should be addressed to M.Z.-G. (email: mz205@cam.ac.uk)

Before implantation, mammalian embryos consist of the embryonic epiblast and two extra-embryonic lineages, the trophectoderm (TE) and the primitive endoderm (PE), organised into the hollow, spherical blastocyst. Upon implantation, this simple structure becomes remodelled[1–5], in the case of the mouse embryo, into the so-called egg cylinder[3]. During this period, the pluripotent epiblast polarises and forms a central lumen[6], while maintaining its position relative to the extra-embryonic tissues.

In the blastocyst, the TE is subdivided into polar TE, which covers the epiblast at the embryonic pole, and mural TE, which overlays the blastocyst cavity at the abembryonic pole[7,8]. During implantation development, the polar TE remains multipotent and gives rise to the extra-embryonic ectoderm (ExE)[9], which will contribute to the embryonic part of the placenta[10]. The mural TE, on the other hand, differentiates into primary trophoblast giant cells, which are necessary for embryo implantation[11,12]. Blastocysts reconstituted by injecting epiblasts into trophospheres formed from isolated mural TE cells, can develop into post-implantation embryos[13–15]. Thus, the distinct behaviour of polar and mural TE has been proposed to result from differential signalling emanating from the epiblast[13–15].

The epiblast expresses FGF ligands during implantation development[16]. It has been suggested that such FGF signalling is necessary for post-implantation TE development because the isolation and maintenance of trophoblast stem cells in culture depends upon FGF4 to sustain its multipotent character[17]. However, in addition to its role in trophoblast stem cell maintenance, FGF signalling is necessary for pre-implantation development[18] as it controls PE specification[19–23]. Specifically, FGF4 expressed by the ICM signals through FGFR1 and FGFR2 to promote and maintain the expression of PE lineage markers[24–26]. Consequently, knockout mice for FGFR1,2 and FGF4 display pre-implantation phenotypes owing to impaired PE specification[18,25–27] and so the role of FGF signalling in TE morphogenesis during implantation development has remained unclear.

Here, we demonstrate that FGF ligands produced by the epiblast are necessary for maintenance of trophectoderm multipotency and the differential fate acquisition of polar and mural trophectoderm, which leads to the establishment of a tissue boundary at the polar/mural trophectoderm interface. We show that the formation of this tissue boundary is indispensable for embryogenesis as its inhibition leads to defective polar trophectoderm expansion into the extra-embryonic ectoderm. Last, we demonstrate that newly formed extra-embryonic ectoderm folds, resulting in cell shape changes driven spreading of the PE. These results demonstrate how morphogenesis of the extra-embryonic tissues drive the remodelling of the mouse embryo during implantation.

## Results

**Trophectoderm cell flow ceases upon implantation.** The pre-implantation blastocyst comprises three distinct tissues spatially distributed along the embryonic–abembryonic (Em–Ab) axis: the Em pole comprises the epiblast surrounded by the polar TE and the PE, whereas the Ab pole consists of the mural TE encompassing the blastocyst cavity (Fig. 1a). Upon implantation, the spherically shaped blastocyst transforms into the egg cylinder. During this process, the polar TE generates the ExE. This is followed by formation of visceral endoderm (VE) from the PE and its spreading to cover both epiblast and ExE (Fig. 1a–c). It is known that before its expansion, the polar TE exhibits a flow towards the mural TE, leading to an increase in mural TE cell population[9,28–30]. As such a polar to mural flow of the TE is not compatible with expansion of the polar TE in the opposite

direction to form the ExE, we hypothesised that the polar TE cell flow must be terminated to allow ExE formation. To test this hypothesis, we filmed developing blastocysts expressing Cdx2-GFP to visualise the TE at two developmental stages (Fig. 1d–i, Supplementary Fig. 1a–c, Supplementary Movie 1–3). Analysis of individual cell behaviour in pre-implantation blastocysts revealed the polar to mural TE flow (Fig. 1d–g, Supplementary Fig. 1a, Supplementary Movie 1), which was terminated in implanting blastocysts (Fig. 1h–j, Supplementary Fig. 1b–c, Supplementary Movie 2–3). Interestingly, we noticed that as the polar to mural TE flow stopped, a gradient of Cdx2, indicative of TE potency[31], became established with higher levels of Cdx2 on the polar side than on the mural side (Supplementary Fig. 1b, Supplementary Movie 3). This gradient of Cdx2 expression was evident only upon termination of the flow and not before implantation when the polar to mural TE flow was taking place (Supplementary Fig. 1d). These results suggest that differential properties of polar and mural TE regulate polar TE cell flow during implantation.

**Trophectoderm tissue boundary forms during implantation.** We next sought to determine how the termination of the polar to mural TE flow is achieved. In various systems, cell populations with different fates and function become separated by the formation of tissue boundaries. These are characterised by high actomyosin contractility at the respective interface of the two populations to prevent cell mixing[32–36]. To investigate whether the differentiation within the TE results in the generation of such a boundary between polar and mural parts, we analysed the distribution of actomyosin during and after termination of the flow. We found high actomyosin contractility at the interface between TE cells expressing Cdx2 at high and low levels (Fig. 2a, b, Supplementary Fig. 2).

As cell–cell junction angles are known to undergo a characteristic widening upon tissue boundary formation[37], we measured the angles between cells at the putative boundary within the TE. This revealed how the angles between cell junctions increased as the entire boundary became linear (Fig. 2c, d, Supplementary Movie 4). These results indicate that a tissue boundary becomes established between polar and mural parts of the TE when the blastocyst initiates its transformation into the egg cylinder.

**Polar trophectoderm expansion follows tissue boundary formation.** To determine the dynamics of polar/mural TE boundary formation and polar TE expansion to form the ExE, we filmed the development of blastocysts recovered from the mother at the time of implantation (E4.5). We observed that polar TE expansion occurred after tissue boundary formation (Fig. 3a, Supplementary Movie 5). In support of this, we observed cell shape changes in the TE once cell–cell junction angles increased. Whereas in early implanting blastocysts, polar TE cells were squamous, they transformed first into cuboidal and then columnar cells exhibiting reduced apical cell area following the polar/mural TE boundary formation (Fig. 3b, c, Supplementary Fig. 3a–b). During the same developmental period, the density of polar TE cells increased (Supplementary Fig. 3c). Upon acquisition of columnar shape, polar TE cells changed the orientation of their division planes (Fig. 3d–f, Supplementary Movie 6). Before tissue boundary formation and for as long as the polar TE epithelium remained single-layered, these cells divided perpendicular to the Em/Ab axis of the embryo and parallel to the epithelial layer (Fig. 3f, Supplementary Fig. 3d, Supplementary Movie 2–4). In contrast, once the tissue boundary formed and the cells had acquired columnar morphology, they divided along their long axes and therefore parallel to the Em/Ab axis and perpendicular to the

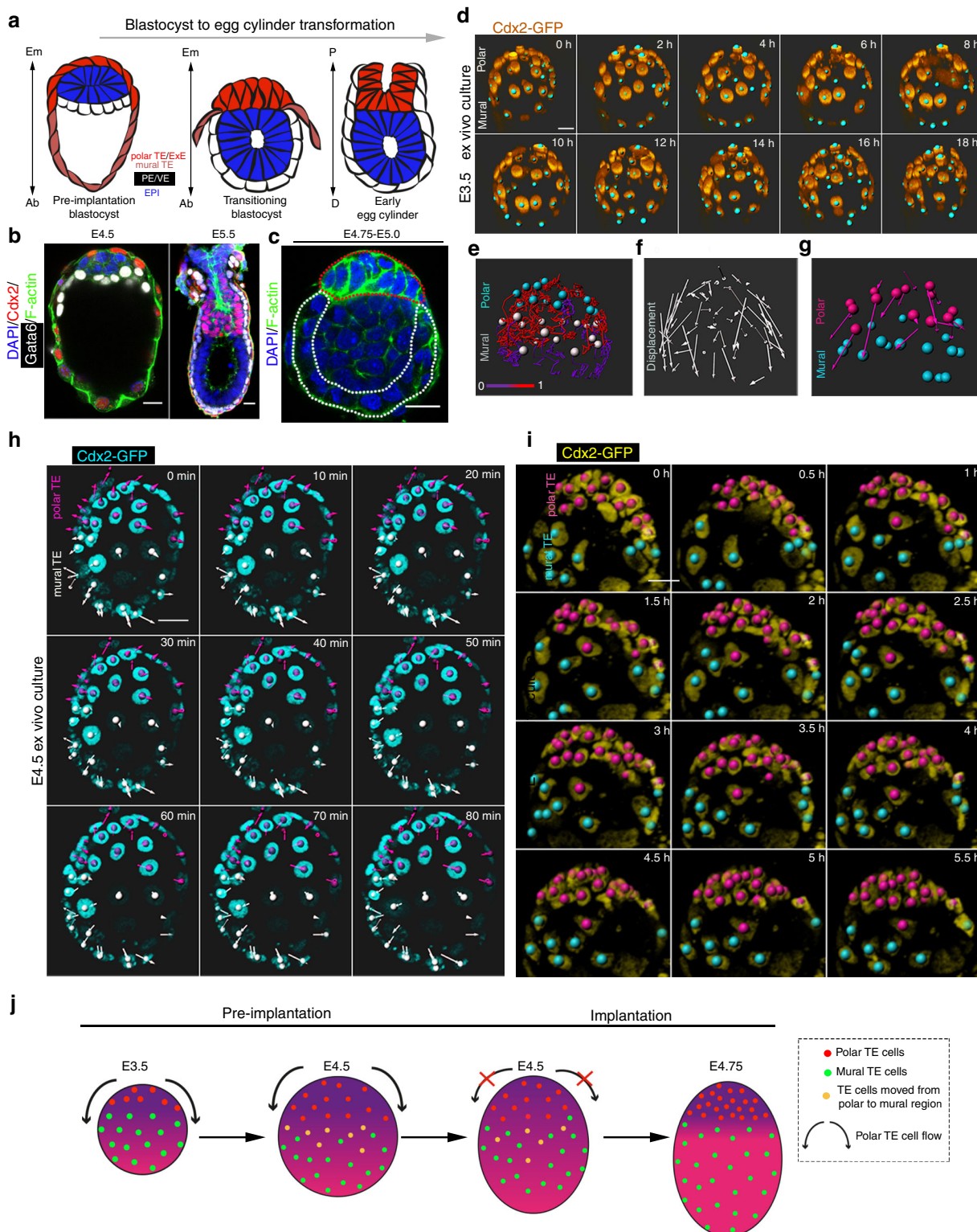

epithelial layer. This resulted in expansion of the polar TE into a multi-layered epithelium (Fig. 3d–f, Supplementary Movie 6). These observations suggest that the formation of a polar/mural TE boundary correlates with expansion of the polar TE into the ExE during egg cylinder morphogenesis.

**FGF and actomyosin control tissue boundary formation.** It has been proposed that positional information provided by the epiblast might be responsible for the differentiation into polar and mural TE, because only the polar but not the mural TE is in direct contact with the epiblast[13,14,38]. FGF signalling is necessary for the maintenance of trophoblast stem cell character and is active in the ExE, which forms from the polar TE in early post-implantation stages[17,27,39]. At the time of implantation, the epiblast expresses FGF ligands[16,40,41], mainly FGF4 and FGF5 (Supplementary Fig. 4a), whereas the TE expresses FGF receptors (FGFR1, 2)[25,42,43]. Therefore, we hypothesised that maintenance

**Fig. 1** Polar trophectoderm transformation. **a** Schematic for blastocyst to egg cylinder transformation. **b** Embryos at late pre-implantation blastocyst (E4.5) and early egg cylinder (E5.0–E5.25) stages stained for markers for primitive endoderm (PE)/visceral endoderm (VE) (Gata6) and trophectoderm (TE)/ extra-embryonic ectoderm (ExE) (Cdx2). Representative of 20 embryos for each stage. **c** Embryo at peri-implantation stage (E4.75–E5.0). Polar trophectoderm (pTE) expansion to form the ExE (red outline) precedes PE spreading (white outlines) during blastocyst to egg cylinder transition. Representative of 20 embryos. **d** Stills from a time lapse movie (Supplementary Movie 1) of a Cdx2-GFP E3.5 blastocyst showing flow of cells from polar to mural TE. Cyan dots: tracking of individual cell nuclei. **e** Colour coded cell tracks according to number of generations as extracted after single cell tracking from Supplementary Movie 1. Cyan dots: polar TE cells. Grey dots: mural TE cells. **f** Displacement map extracted after single cell tracking from Supplementary Movie 1. **g** Polar TE displacement map extracted after single cell tracking from Supplementary Movie 1. **h** Stills from a time lapse movie of Cdx2-GFP E4.5 implanting blastocyst combined with the displacement map (magenta: polar TE; white: mural TE) showing stop of cell flow from polar to mural TE upon implantation. **i** Stills from a time lapse movie of Cdx2-GFP E4.5 implanting blastocyst combined with single cell tracking (magenta: polar TE; cyan: mural TE) showing stop of cell flow from polar to mural TE upon implantation. **j** Model of polar TE cell flow in pre- and post implantation blastocysts. Scale bars = 20 µm

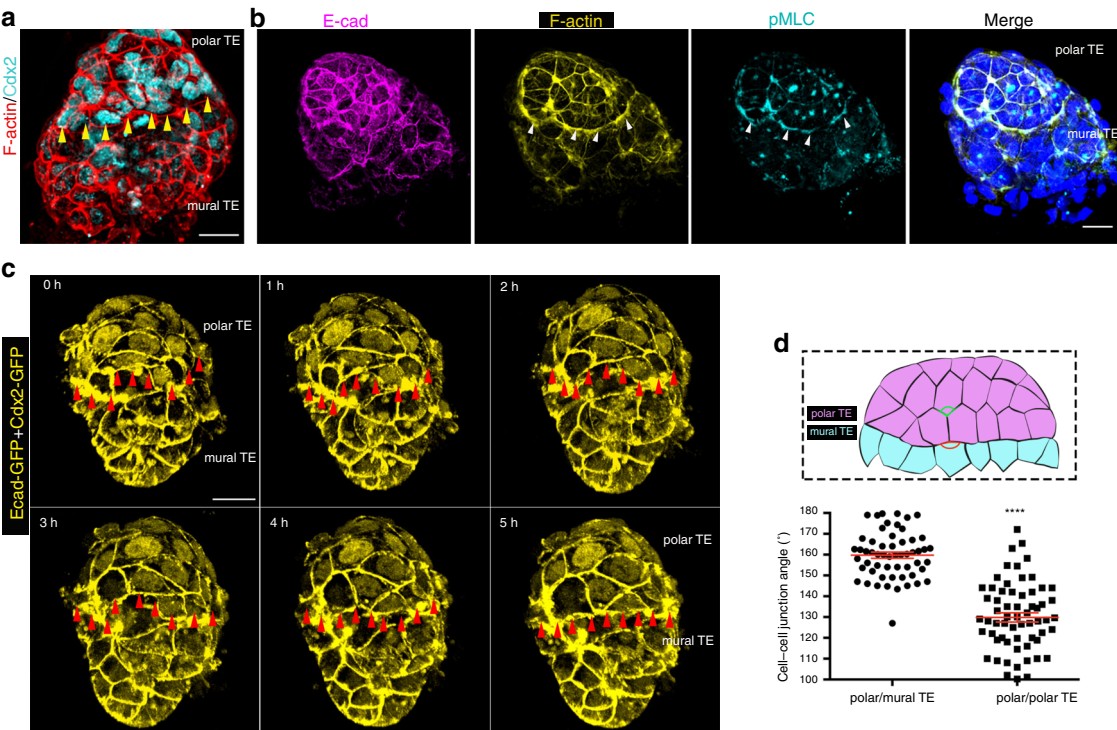

**Fig. 2** Establishment of tissue boundary within the trophectoderm. **a** Maximum intensity projection (MIP) image of an implanted E4.75 blastocyst. Arrowheads indicate tissue boundary between polar and mural TE. Representative example of 20 embryos. **b** MIP image of an implanted E4.75 blastocyst. Arrowheads indicate actomyosin enrichment at the boundary between polar and mural TE. Representative example of eight embryos. **c** Stills from a time lapse movie of Cdx2/E-Cad GFP E4.5-implanting blastocyst. Red line follows the boundary between high-Cdx2 cells (polar TE) and low-Cdx2 cells (mural TE). As development progresses, polar and mural TE cells segregate and a linear tissue boundary formed (arrowheads). **d** Quantification of cell–cell junction angle at the polar/mural TE boundary and within polar TE. Two-sided unpaired student's $t$ test; ****$P < 0.0001$; mean ± SEM. $n = 56$ polar/mural TE and 63 polar TE cell junctions. Source data are provided as a Source Data file. Scale bars = 20 µm

of Cdx2 expression in the polar TE at the time of implantation could be a response of the tissue to FGF signalling. To test this possibility, we first assessed the status of FGF signalling before and during implantation using pERK and pAKT as a read out of its activity[39,44]. This revealed that FGF signalling was not active in the polar TE before implantation but showed restricted activation in the polar TE in implanting blastocysts (Fig. 4a, b, Supplementary Fig. 4b–c).

To examine whether FGF signalling is indeed responsible for the maintenance of multipotency in the polar TE at implantation stages, we inhibited FGF signalling in ex vivo cultured blastocysts through treatment with a specific FGF receptor inhibitor[39] for 20 h (Fig. 4c). This treatment gave us temporal control over FGF inhibition and avoided an effect on the earlier specification of PE[18,21,25–27]. Inhibition of FGF signalling resulted in the loss of Cdx2 expression in the polar TE (Supplementary Fig. 4d) without affecting specification of PE, which was already formed by the

time of treatment. To confirm the role of FGF in TE multipotency maintenance, we next recovered blastocysts at E3.5 and cultured them for 24 h to the E4.5 stage before treating them with FGF for 24 h. This revealed that the presence of FGF prevents mural TE differentiation as indicated by Cdx2 expression levels (Supplementary Fig. 5a–c). These results indicate that positional information from the epiblast in the form of FGF signalling is responsible for the maintenance of multipotency in the polar TE, evident from the appearance of a gradient in Cdx2 expression.

As we observed high actomyosin contractility at the interface of polar/mural TE and as FGF signalling is responsible for multipotency maintenance in the polar TE resulting in the appearance of a Cdx2 expression gradient in the TE, we next wished to determine whether actomyosin contractility and FGF signalling have a role in the formation of the boundary between polar and mural TE. To this end, we cultured E4.5 blastocysts for 20 h in the presence of ROCK inhibitor or blebbistatin(−) to

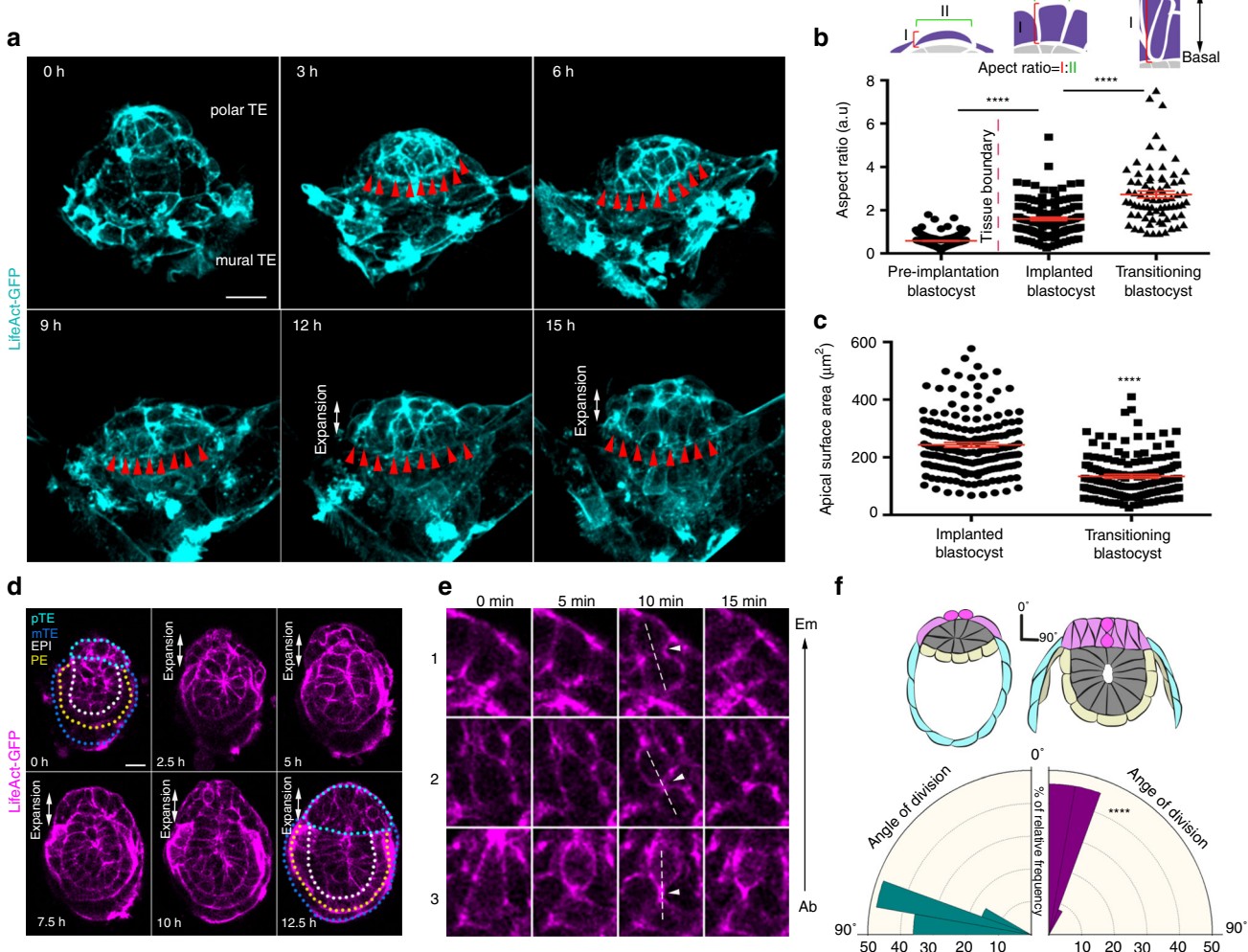

**Fig. 3** Cell shape changes and polarized cell divisions drive polar TE expansion. **a** Stills from a time lapse movie of Lifeact-GFP blastocyst. Red arrowheads indicate the polar/mural TE boundary. Boundary formation is followed by polar TE expansion. **b** Quantification of polar TE cell aspect ratio at different developmental stages. Two-sided unpaired student's $t$ test; ****$P < 0.0001$; mean ± SEM. Pre-implantation blastocyst: $n = 216$ cells; Implanted blastocyst: $n = 152$ cells; Transitioning blastocyst; $n = 75$ cells. **c** Quantification of polar TE apical cell surface are. Two-sided unpaired student's $t$ test; ****$P < 0.0001$; mean ± SEM. Blastocyst before: $n = 186$ cells; Transitioning blastocyst; $n = 163$ cells. **d** Stills from a time lapse movie of Lifeact-GFP implanted blastocyst. Polar TE expansion is evident at 2.5 h upon polar TE cell shape changes and polarised cell division. **e** Representative examples of polar TE cells dividing parallel to Embryonic(Em)/Abembryonic(Ab) axis during polar TE expansion. **f** Rose diagram for quantification of cell division orientation in blastocysts before (green) and during (purple) polar TE expansion. Kolmogorov–Smirnov test: ****$p < 0.0001$. Blastocyst before polar TE expansion: $n = 24$, Bastocyst during polar TE expansion: $n = 36$. Source data are provided as a Source Data file. Scale bars = 20 μm

inhibit actomyosin contractility[45] (Fig. 4c, Supplementary Fig. 6a). Inhibition of actomyosin contractility resulted in significantly narrowed angles of the cell–cell borders at the polar/mural TE interface indicative of defective boundary formation (Fig. 4d–f, Supplementary Fig. 6b–c), thus confirming the essential role of actomyosin contractility in this process. Similarly, inhibition of FGF signalling during the same developmental time window led to defective polar/mural TE boundary formation (Fig. 4d–f). Together, these results indicate the necessity of FGF signalling from the epiblast to the polar TE for the formation of the tissue boundary between polar and mural TE.

To understand the role of the mural/polar TE boundary in subsequent development, we perturbed its formation by inhibiting either FGF signalling or actomyosin contractility in implanting embryos and following their subsequent development. We found that inhibition of FGF signalling or actomyosin contractility led to failure of both the polar TE to expand to form the ExE and egg cylinder formation (Fig. 4g, h, Supplementary Fig. 6b, d).

These results indicate that formation of the polar/mural TE tissue boundary is a key step in blastocyst to egg cylinder morphogenesis (Fig. 4i).

**PE morphogenesis**. During the final stage of the blastocyst to egg cylinder transformation, the PE differentiates into the VE, which will cover both the epiblast and the newly formed ExE (Fig. 1a–c). To gain insight into this process, we analysed the behaviour of the PE at consecutive stages of implantation development in vivo and in vitro. We found that PE cells localised at the distal tip of the epiblast facing the blastocyst cavity displayed a cuboidal shape, which became columnar upon implantation while still confined to the distal tip (Fig. 5a, b, Supplementary Fig. 7a–b, Supplementary Fig. 8, Supplementary Movie 7). However, as development progressed, PE cells changed shape again to become cuboidal, concomitant with their spreading over the epiblast and the expansion of the polar TE (Fig. 5a–c, Supplementary Fig. 7b, Supplementary Fig. 8–9, Supplementary Movie 7). These results suggest that

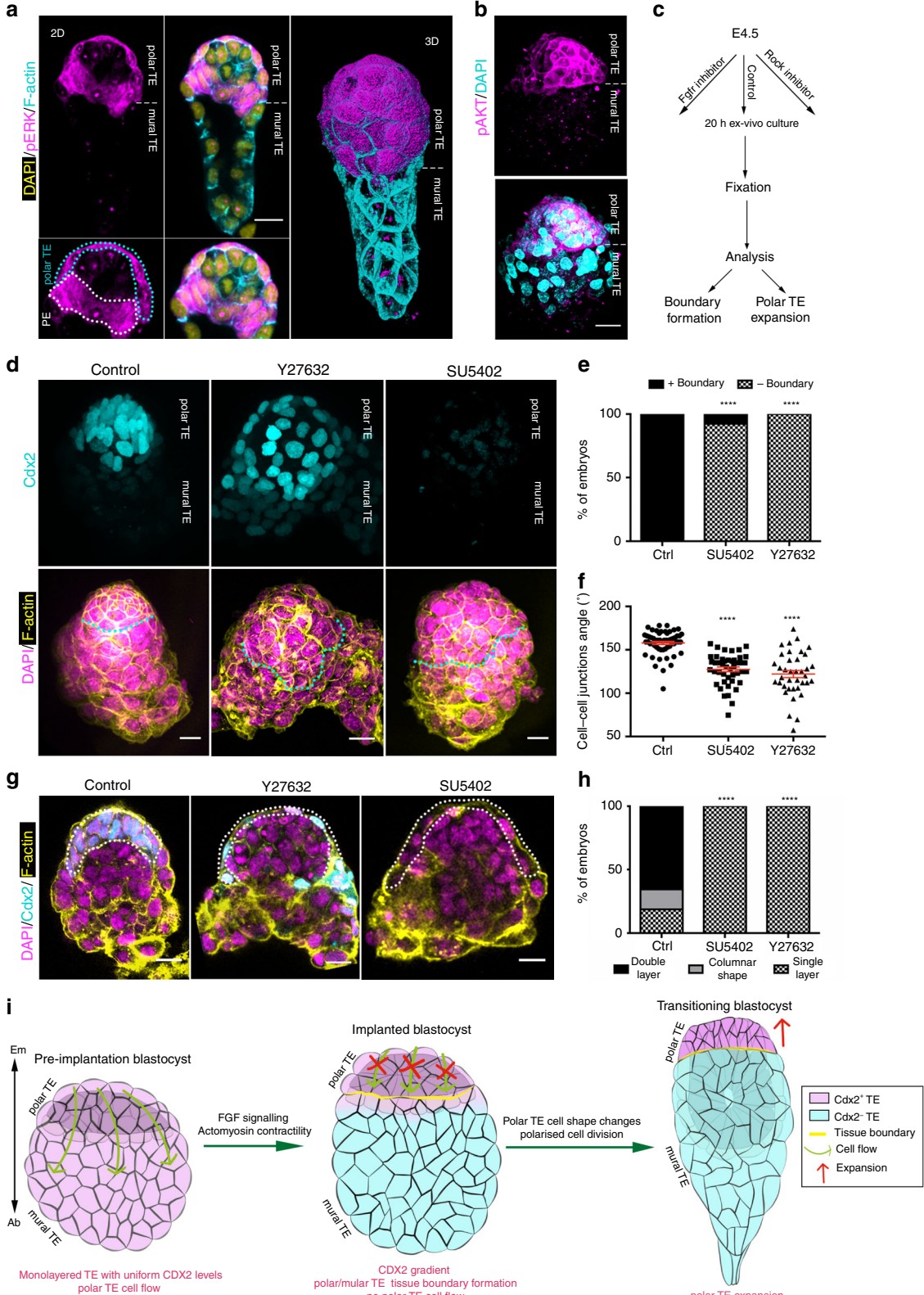

expansion of the polar TE and spreading of the PE/VE are coordinated events.

During implantation development, the PE gives rise not only to the VE but also to the parietal endoderm[46]. The parietal endoderm migrates toward the mural TE with Reichert's membrane acting as a substrate for its migration[47]. As it is possible that parietal endoderm migration might also contribute to PE morphogenesis during implantation development, we examined the movement of parietal endoderm cells in implanting blastocysts (E4.5–E4.75). We found that parietal endoderm migration is completed before the TE expands and the PE spreads over the epiblast (Fig. 5d). These results strongly suggest that parietal endoderm migration is not involved in PE/VE morphogenesis.

**Fig. 4** Tissue boundary is necessary for polar trophectoderm expansion. **a** Implanted blastocyst (E4.75) stained for phosphorylated ERK (pERK). pERK is detected in the primitive endoderm (PE) and the trophectoderm (TE). pERK is restricted to the polar region of the TE. **b** Implanted blastocyst (E4.75) stained for phosphorylated AKT (pAKT). pAKT is restricted to the polar region of the TE. For **a** and **b**, representative example of 10 embryos. **c** Schematic representation of the experimental design to examine the contribution of actomyosin contractility and FGF signalling in the formation of polar/mural TE tissue boundary formation. **d** Representative examples of control, Y27632 (ROCK inhibitor) and SU5402 (Fgfr inhibitor) treated blastocysts cultured as described in **c** and analysed for the presence of polar/mural TE boundary (cyan dotted line). Polar/mural tissue boundary formation is defective in the absence of actomyosin contractility and FGF signalling. **e** Quantification of polar/mural TE boundary formation efficiency in control, Y27632 and SU5402-treated blastocysts. $\chi^2$ test; ****$P < 0.0001$, mean ± SEM. For **d** and **e** FGF $n = 29$ control, 15 Y27632 treated and 15 SU5402-treated embryos. **f** Quantification of cell–cell junction angles at the polar/mural TE interface. The angles of junctions after abrogation of actomyosin contractility (Y27632) and signalling (SU5402) are narrowed in agreement with the defective formation of polar/mural TE tissue boundary in these conditions. Two-sided unpaired student's $t$ test; ****$P < 0.0001$; mean ± SEM. Control: $n = 57$; Y27632: $n = 38$; SU5402: $n = 45$. **g** Representative examples of control, Y27632 (ROCK inhibitor) and SU5402 (FGFR inhibitor)-treated blastocyst cultured as described in **b** and analysed for polar TE (white outline) expansion. **h** Quantification polar TE expansion efficiency in control, Y27632, and SU5402-treated blastocysts. $\chi^2$ test; ****$P < 0.0001$. For **f** and **g** $n = 29$ control, 15 Y27632 treated and 15 SU5402-treated embryos. **i** Model for the molecular and cellular events necessary for polar TE expansion during peri-implantation morphogenesis. Source data are provided as a Source Data file. Scale bars = 20 μm

Immediately preceding egg cylinder formation, the PE covers only the epiblast and is not in contact with the polar TE (Fig. 5a, b, Supplementary Figs. 8, 9). We hypothesised that the polar TE expansion into ExE might act as a substrate for migration of the PE to cover the ExE. This would require degradation of the Reichert's membrane that encloses the distal half of the embryo forming a continuous membrane with the embryonic basement membrane (Fig. 5e upper panel). However, we found that after spreading of the PE, the whole egg cylinder remained enclosed by the continuous embryonic and Reichert's basement membrane (Fig. 5e lower panel). Together, these observations suggest that active migration of PE cells is not required for egg cylinder formation.

**Trophectoderm folding results in egg cylinder formation**. To understand how the PE develops into the VE, which covers the entire embryo during egg cylinder formation, we performed live imaging of Lifeact-GFP transgenic embryos from an early post-implantation stage (E4.75). Analyses of our movies revealed that the VE spreads over the ExE concomitantly with the folding of the ExE (Fig. 6a–f, Supplementary Figs. 8–10, Supplementary Movie 8–9). Examination of embryos developing in vivo and fixed immediately upon recovery demonstrated that ExE folding is followed by directional growth of the embryo towards the Ab pole of the blastocyst cavity, which results in filling of the previously hollow space (Supplementary Fig. 9).

Based on the above observations we hypothesised that ExE folding could be the driving force for VE morphogenesis. Specifically, we considered that ExE folding and embryo movement toward the Ab pole might lead to the development of stretching forces leading to cell shape-driven spreading of the VE[48]. To test this hypothesis, we measured the orientation of the long axes of VE cells and found that, upon ExE folding, the most proximal cells of the VE became aligned with the proximo-distal axis of the embryo (Fig. 6g, Supplementary Fig. 9). These results suggest the possibility that the VE responds passively to tensile forces[48–50] generated along the proximal-distal axis as a result of ExE invagination.

To test this possibility further, we sought to prevent the ExE folding, which requires apical constriction[51] and so can be prevented using a ROCK inhibitor[45]. We found that ROCK inhibition applied from an early post-implantation stage (E4.75), resulted in failure of ExE invagination, VE spreading, and consequently egg cylinder formation (Fig. 6h, i, Supplementary Fig. 11a, Supplementary Movie 10). To exclude any contribution of active VE cell movement as it develops to cover the ExE, we inhibited Rac, which is indispensable for collective cell migration[52,53], and filmed the subsequent development.

Inhibition of Rac affected neither expansion of the VE to cover the epiblast and ExE nor egg cylinder formation (Supplementary Fig. 11b), in accord with the phenotype of embryos that are Rac1 knockout only in the VE[52]. Thus, the expansion of the VE appears to be a passive response to egg cylinder formation.

## Discussion

In this study, we describe morphogenetic events driving the critical phase of mammalian embryo development that takes place during implantation. Our findings demonstrate an interplay between positional and chemical signals as well as physical cellular responses that leads to global tissue rearrangements as the mouse blastocyst transforms into the egg cylinder. Specifically, we describe how FGF signalling from the epiblast determines the fate of a subset of TE cells and subsequent morphogenesis that generates forces necessary for the spreading of the PE/VE to cover not only the epiblast but also the expanding ExE (Fig. 6j).

Here, we show that the first morphogenetic event during the blastocyst to egg cylinder transition is the formation of a tissue boundary between the polar and mural TE. Formation of this boundary prevents the flow of polar TE cells toward the mural region that occurs at pre-implantation stages. Abrogation of the formation of the TE boundary shows that it is necessary for proper embryo development. Our results show that actomyosin contractility at the interface between the mural and polar TE is indispensable for tissue boundary formation. Such an increase in actomyosin contractility at the interface of differentially fated cells is a feature of tissue boundary formation in various model systems[54]. In the case of the polar/mural TE boundary, the localised increase of actomyosin contractility at the interface could either be a result of heterophilic adhesions[32,37] or repulsive cues generated by Ephrin/Eph signalling[55,56].

Numerous studies have described the role of FGF signalling in PE specification and in the maintenance of trophoblast stem cells[17,18,21,25–27,41]. However how FGF signalling instructs the peri-implantation morphogenesis of the TE has not previously been studied owing to the pre-implantation phenotypes of FGF and FGFR knockout embryos[18,25,26]. Here, we show that the TE tissue boundary forms in response to FGF signalling originating from the epiblast. Specifically, FGF ligands produced by the epiblast act as positional cues driving differential fate acquisition of the polar (undifferentiated) and mural (differentiated) TE. Importantly, whereas FGF signalling is indispensable for polar TE morphogenesis and multipotency maintenance upon implantation, it is dispensable for maintenance of the TE's multipotent state in pre-implantation stages[25,41]. During pre-implantation development, TE multipotency is maintained by co-operation of Notch and Hippo pathways[57–59]. It will be important to identify

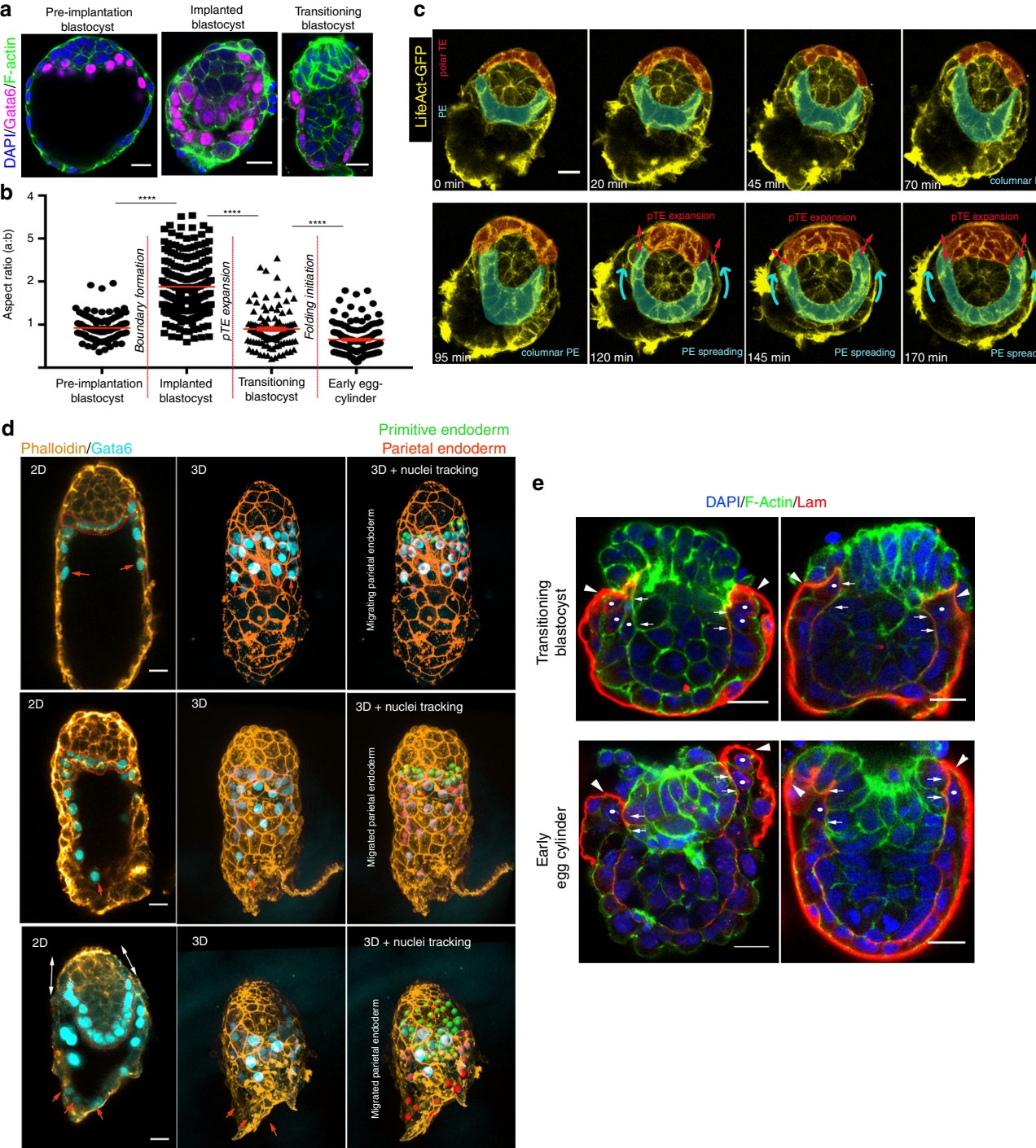

**Fig. 5** Primitive endoderm behaviour during the blastocyst to egg cylinder transition. **a** Examples of embryos used for analysis of primitive endoderm (PE) cell shape at different stages of peri-implantation development. Representative of 20 embryos for each stage. **b** Quantification of PE cell aspect ratio at different developmental stages. The red lines between different columns correspond to events taking place within the trophectoderm (TE). Two-sided unpaired student's *t* test; ****$P < 0.0001$; mean ± SEM. Pre-implantation blastocyst: $n = 117$ cells; Implanted blastocyst: $n = 219$ cells; Transitioning blastocyst: $n = 117$ cells; early egg cylinder: $n = 184$ cells. **c** Stills from a time lapse movie of Lifeact-GFP E4.5-implanting blastocyst. PE (cyan) acquires a columnar morphology at implantation stages and this is followed by tissue spreading through cell-shape changes upon polar TE (red) expansion as quantified in **b**. Red double-headed arrows: polar TE expansion. Cyan arrows: PE spreading. Stills without coloured overlay are presented in Supplementary Fig. 7b. **d** Representative examples of implanting blastocyst (E4.5–E4.75). Parietal endoderm migration is completed before the expansion of the TE (double-headed arrows in bottom row) and before PE (red outline in 2D images) spreading over the lateral sides of the epiblast (bottom row). Red arrows indicate the front of the migrating parietal endoderm. **e** Representative examples of transitioning blastocysts (E5.0) displaying expanded polar trophectoderm and early egg cylinders. Primitive endoderm cells are found enclosed between the embryonic and Reichert's basement membrane during blastocyst to egg cylinder transformation. arrowheads: Reichert's membrane. Arrows: embryonic basement membrane. Dots: proximal cells. Representative images of 20 embryos per stage. Source data are provided as a Source Data file. Scale bars = 20 μm

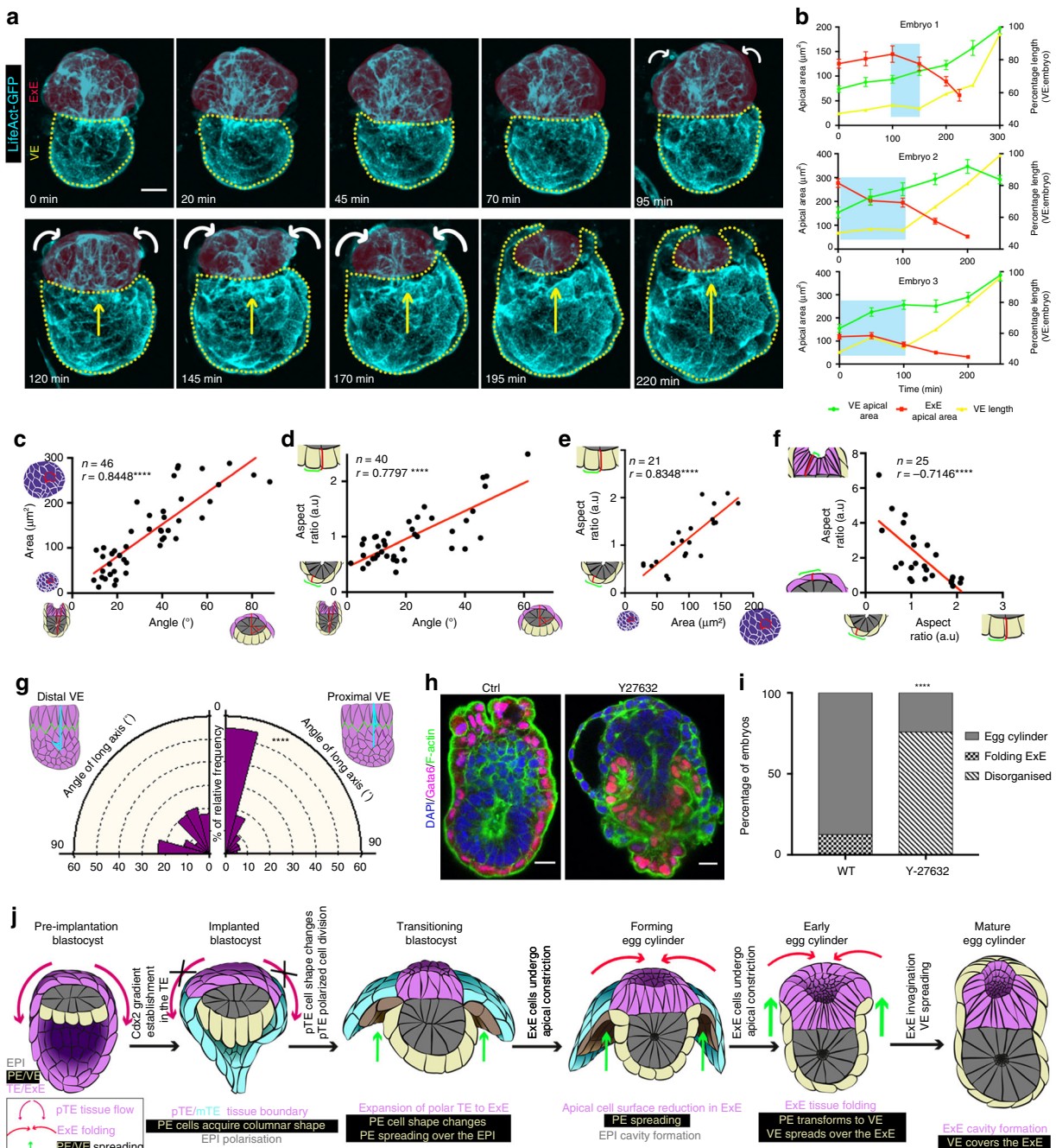

**Fig. 6** Trophectoderm folding orchestrates the final step the blastocyst to egg cylinder transformation. **a** Stills from a time lapse movie of Lifeact-GFP blastocyst. ExE tissue (red) folding (white arrows) through apical constriction is followed by spreading (yellow arrows) of the visceral endoderm (VE) (yellow outline) and formation of egg cylinder. Stills without coloured overlay are presented in Supplementary Fig. 10a. **b** Quantification of ExE apical cell surface area, VE apical cell surface area and VE length through time during blastocyst to egg cylinder transformation from three individual embryos shown in Supplementary Movie 8. Cyan boxes indicate the developmental period, during which ExE cell's apical constriction precedes VE spreading. VE tissue spreading during egg cylinder formation is owing to cell-shape changes as quantified in Fig. 5b. **c** Quantification of polar TE/ExE apical cell surface area relative to PE/VE position. **d** Quantification of PE/VE cell aspect ratio relative to PE/VE position. **e** Quantification of PE/VE cell aspect ratio relative to polar TE/ExE apical cell surface area. **f** Quantification of polar TE/ExE cell aspect ratio relative to PE/VE cell aspect ratio. For **c–f**, Pearson correlation test was used, ****p < 0.0001. **g** Rose diagram for quantification of long cell axis orientation in distal and proximal VE at the early egg cylinder stage. Kolmogorov–Smirnov test: ****p < 0.0001. Distal VE: n = 153 cells, Proximal VE: n = 75 cells. **h** Representative examples of control and Y27632 (ROCK inhibitor)-treated embryos. Embryos were recovered at E4.75 and cultured ex vivo for 12 hours. In the absence of actomyosin contractility (Y27632) egg cylinder formation fails due to defective ExE folding (Supplementary Movie 10). **i** Quantification of blastocyst to egg cylinder transformation efficiency in control, and Y27632 treated embryos. $\chi^2$ test; ****P < 0.0001. For **h** and **i** n = 16 control and 29 Y27632-treated embryos. **j** Model of blastocyst to egg cylinder transformation as a result of TE morphogenesis. Source data are provided as a Source Data file. Scale bars = 20 μm

the mechanisms controlling the switch from FGF-independent to FGF-dependent maintenance of TE multipotency during peri-implantation development in future studies.

Upon TE tissue boundary formation, polar TE cells change from squamous to columnar morphologies. Similar shape changes from squamous to columnar morphologies are evident in the PE during egg cylinder formation. This could result from cell crowding owing to continuous cell proliferation in a restricted environment[48,60]. Polar TE cells divide in a confined area imposed upon them by the formation of the tissue boundary between themselves and the mural TE. The PE tissue increases its cell number while being confined to the distal site of the epiblast. Polar TE cell shape changes are followed by polarised cell divisions that both contribute to the formation of the double cell layer of the ExE. TE expansion is followed by PE spreading over the epiblast, during which the PE cells change their shape from columnar to cuboidal. These cell shape changes could be a result of stretching forces experienced by the PE during TE expansion and the movement of the epiblast towards the abembryonic pole of the implanting blastocyst[48].

The final step of egg cylinder formation is orchestrated by invagination of the newly formed ExE, which is driven by apical constriction of ExE cells[51]. Apical constriction as a mechanism for tissue folding has been described in different models[61–64]. Here, folding is initiated by apical constriction only when the polar TE expands to form the ExE. The proliferation of the polar TE during this process leads to an increased cell density, which correlates with an increase in pressure forces within epithelial tissues[65]. It has been reported that mechanical signals regulate myosin II distribution and turnover during tissue morphogenesis and wound healing[66–68]. Thus, we hypothesise that initiation of ExE folding as a result of apical constriction might be regulated by mechanically mediated enrichment of myosin II through a mechano-transductive cascade.

Our results demonstrate that ExE folding is followed by VE spreading over the ExE, which is necessary for the completion of the blastocyst to egg cylinder transition. We find that neither parietal endoderm migration nor active PE cell migration contribute to this process. Just as with the spreading of PE over the epiblast, the spreading of VE over the ExE is probably driven by mechanically induced cell shape changes[48]. In this case, the force generating mechanism is the folding of the ExE, which results in the directional growth of the embryo towards the Ab pole[9]. In agreement, prevention of ExE folding by inhibition of apical constriction results in impaired egg cylinder formation and VE spreading.

In summary, our results demonstrate that TE morphogenesis orchestrates the shape acquisition of the post-implantation mouse embryo during the blastocyst to egg cylinder transition. The fact that murine and primate embryos are similar in morphology to the time of implantation, when they begin to display differential behaviour of the TE[1–5], suggests that the different shapes of murine and primate post-implantation embryos might be attributed to TE peri-implantation morphogenesis. Future studies on TE morphogenesis in different species should shed light onto the role of the TE in the mechanisms that lead to the characteristic shapes of implantation embryos of different mammalian species.

## Methods

**Embryo recovery and culture.** In accordance with national and international guidelines, the mice used were kept in the animal facility. All experiments performed have been regulated by the Animals (Scientific Procedures) Act 1986 Amendment Regulations 2012 and additional ethical review by the University of Cambridge Animal Welfare and Ethical Review Body (AWERB). Experiments were authorised by the Home Office (Licence number: 70/8864). Upon any identification of health concern, mice were culled by cervical dislocation. Pre-implantation stage embryos

from Cdx2-GFP transgenic or CD1 females mated with either Cdx2-GFP transgenic, F1, MF1, or CD1 males were flushed from the uteri and cultured in KSOM. Peri-implantation and early post-implantation embryos from Pdgfra-GFP, Lifeact-GFP, E-Cadherin-GFP, Cdx2-GFP, and CD1 females mated with either Pdgfra-GFP, LIFE-ACTT-GFP, E-Cadherin-GFP, Cdx2-GFP, F1, MF1, and CD1 males were dissected from the uteri or deciduas and cultured in drops of Advanced DMEM/F12 (GIBCO) containing 20% fetal bovine serum and supplemented with 2 mM L-glutamine (GIBCO), 1 mM sodium pyruvate (GIBCO), penicillin (25 units/ml)/streptomycin (25 μg/ml) (GIBCO), 1 × ITS-X (Invitrogen). The medium drops were covered with mineral oil and the embryos were cultured at 37 °C and 5% $CO_2$.

For ROCK inhibition, the embryos were placed in drops of medium, either containing Y27632 ROCK inhibitor (100 μM) dissolved in dimethyl sulfoxide (DMSO) or solely DMSO as a control and incubated 20 h for examination of tissue boundary formation and 12 h for examination of ExE folding at the above described conditions. For blebbistatin(−) treatment, the embryos were placed in drops of medium, either containing 50 μM blebbistatin(−) dissolved in DMSO or solely DMSO as a control and incubated 20 h for examination of tissue boundary formation at the above described conditions. For FGF treatment, embryos were recovered at E3.5 cultured for 24 h in KSOM and then cultured for another 24 h in IVC medium in the presence or the absence of 1000 ng/ml Fgf4 + 1 μg/ml Heparin FGFR inhibition was achieved through incubation of blastocysts in medium containing SU5402 at 40 μM or DMSO for the control group. For Rac inhibition, Rac1 Inhibitor II, Z62954982 at a concentration of 100 μM was used and embryos for the control group were incubated in the same amount of DMSO.

**Generation of endogenous E-Cadherin-GFP transgenic mouse line.** A Cdh1-EGFP allele was created by inserting a loxP-stop-loxP-EGFP cassette immediately prior to the endogenous stop codon in the final exon of the Cdh1 gene (ENSMUSE00000469626; Chromosome 8: 106,668,613). The allele was designed that following Cre recombination the stop codon was removed and the EGFP open-reading frame was fused to the coding sequence of the Cdh1 gene creating a Cdh1-EGFP fusion protein. Prior to Cre recombination the loxP sites are separated by 300 bp spacer sequence composed of non-coding sequence from the human Pgk1 gene.

The targeting vector was assembled as follows. Two synthetic DNA fragments (MWG Eurofins) were ligated to form a combined fragment comprising 5′ (300 bp) and 3′ (500 bp) Cdh1 homology arms flanking a loxP-STOP-loxP element immediately adjacent to the last amino-acid codon of Cdh1, an EGFP open reading frame and a FRT site within the sequence of the Cdh1 3′-UTR.

An FRT-Hygro-R cassette was inserted into the above plasmid by co-transfection into EL250 E. coli[69], which express Flp recombinase under arabinose induction.

The DNA fragment bound by the homology arms was excised and recombined[69] into a pool of mouse genomic DNA BAC clones (Source Biosciences) carrying the mouse Cdh1 gene in EL250 E. coli[70].

A retrieval plasmid was generated by cloning a retrieval fragment (comprising 540 bp and 490 bp homology arms separated by a SwaI site) into NotI-AscI linearised pFlexDTA (a modified version of pFlexible)[71]. Following linearization by SwaI, the modified sequences were retrieved from BAC clones in EL250 E. coli by recombining. The retrieved plasmid represents the targeting vector with ~ 5 kb and 4.5 kb homology arms.

The targeting vector was linearised by AscI digestion and transfected into HM1 mESCs[72]. Cells were selected under hygromycin (150 μg/ml) and surviving colonies picked and screened for targeting by long range PCR (using the Roche Expand Long Template PCR System) from within the STOP element or hygromycin-resistance cassette to sequences beyond the ends of the homology arms. Oligo sequences used to screen cells to ensure appropriate targeting of the Cdh1 gene were TACCACGTGGGATACCCCAAG and AGGTATGCCAGAAGCCACAG for the 5′ side and GTCCGAGGGCAAAGGAATAG and GAAGCCTCAGGTTTG GTTCC for the 3′ side.

Following identification of correctly targeted clones, mouse lines were derived by injection of ES cells into C57BL/6 J blastocysts. After breeding of chimeras, germline offspring were identified by coat colour and the presence of the modified allele was confirmed with the 3′ loxP primers described above. Mice were subsequently crossed with a mouse line expressing Flpe (Tg(ACTFLPe)9205Dym) to delete the selectable marker by recombination at the FRT sites[73].

**Immunostaining.** Embryos were fixed directly following dissection in 4% paraformaldehyde in phosphate-buffered saline (PBS) for 20 min at room temperature (RT). Incubation except fixation was carried out in wells coated with filtered fetal calf serum (FCS) to avoid attachment of peri-implantation stages at the bottom of the wells. Permeabilisation was carried out for 20 min at RT through incubation in 0.3% Triton X-100/0.1 M Glycin in PBS. Primary antibodies were prepared in blocking solution (0.1% Tween-20/10% filtered FCS in PBS) and incubated at 4 °C overnight. Secondary antibody incubation was performed at RT for 3 h and carried out in blocking solution as well. Nuclear staining was incubated simultaneously with secondary antibodies using DAPI (10 mg/ml). All washes were carried out in PBS supplemented with 0.1% Tween-20. For pERK staining, embryos were fixed in Methanol and incubated for 15 min at − 20 °C in ice-cold Methanol. Following fixation, the embryos were stained as described above. Primary antibodies used:

Cdx2 (mouse monoclonal, 1:300, Launch Diagnostics, MU392-UC (Biogenex); E-Cadherin (rat, 1:200, Thermo Fisher Scientific, 13-1900); Gata6 (goat, 1:200, R&D Systems, AF1700,); Laminin (rabbit, 1:300, Sigma, L9393); pAKT (rabbit, 1:100, Cell Signaling Technology, 4060 T), pERK (rabbit, 1:100, New England Biolabs, 4377 T); phospho-Myosin Light chain II (rabbit, 1:100, Cell Signaling Technology, 3671 S), podocalyxin (1:300; R&D systems, MAB1556).

Secondary antibodies used:

Alexa Fluor® 488 Phalloidin (1:500, Thermo Fisher Scientific, A12379); Alexa Fluor® 594 Phalloidin (1:500, Thermo Fisher Scientific, A12381); Goat anti-Rat IgG (H + L) Secondary Antibody, Alexa Fluor 647 (donkey, 1:500, Thermo Fisher Scientific, A-21247); Alexa Fluor® 568 Donkey Anti-mouse IgG (donkey, 1:500, Thermo Fisher Scientific, A10037); Donkey anti-Mouse IgG (H + L) Secondary Antibody, Alexa Fluor® 488 (donkey, 1:500, Thermo Fisher Scientific, A-21202); Alexa Fluor 568 Donkey Anti-Rabbit IgG Antibody (donkey, 1:500, Thermo Fisher Scientific, A10042); Alexa Fluor® 647 Donkey Anti-Rabbit IgG (H + L) Antibody (donkey, 1:500, Thermo Fisher Scientific, A-31573); Alexa Fluor 647 Donkey Anti-Mouse IgG (H L) Antibody (donkey, 1:500, Thermo Fisher Scientific; A31571); Donkey anti-Rat IgG (H + L) Secondary Antibody, Alexa Fluor® 488 conjugate (donkey, 1:500, Thermo Fisher Scientific, A-21208)

**Imaging**. Fixed embryos were imaged on a Leica SP8 microscope using a ×63 oil objective. Pre-implantation live blastocysts were imaged on a Spinning Disc ×40 objective at a z-step size of 1 µm. Implanting blastocysts as well as early post-implantation embryos were imaged on a Leica multiphoton SP8 microscope using a ×25 water objective with a z-step size of 1 µm. For excitation, a 488 nm laser was used. Embryos transgenic for Cdx2-GFP, E-Cadherin-GFP, and Pdgfra-GFP were imaged with 1.5% of 488 nm laser. Embryos expressing LifeAct-GFP were imaged at 0.25% of 488 nm laser.

A detailed list of the reagents and equipment used for embryo mounting and culture to optimise live imaging is provided in the supplementary methods. A detailed protocol is uploaded in Nature protocol exchange (ref. [74]).

**Image processing and analysis**. Image analysis of fixed embryos was carried out using the Leica 3D viewer and Fiji. For single cell tracking, the Imaris image analysis software was used. Image processing was performed with the Fiji image processing software.

**Statistics and reproducibility**. GraphPad Prism 6.0 software was used for all statistical analysis performed on the results obtained. For treatments, the embryos were allocated randomly to treatment and control group, but investigators were not blinded to group allocation. The sample size of the experiments carried out was defined based on previous experimental experience. Quantitative data presented, shows the mean ± s.e.m., percentages, or the total number of datapoints obtained. The statistical tests carried out on the quantitative data obtained are annotated in each figure legend. Except annotated otherwise, each experiment shown was carried out at least three times.

**Reporting summary**. Further information on research design is available in the Nature Research Reporting Summary linked to this article.

## Data availability

The authors declare that all data supporting the findings of this study are available within the article and its supplementary information files or from the corresponding author upon reasonable request. The source data of Figs. 1d, 1h, 1i, 2c, 3a, 3d, 5c, 6a, and Supplementary Figs. S1b, S7a, S10c–d, S11a can be found as supplementary movies. The complete data supporting the results discussed in this study are available upon a reasonable request from the corresponding author. The source data underlying Figs. 2d, 3b, 3c, 3f, 4f, 5b, 6b–g, and Supplementary Figs. 3c, 5b, and 10c are provided as a Source Data file.

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

## Acknowledgements

We are grateful to D. Glover, M. Shahbazi, M. Zhu, and C. Kyprianou for feedback on the manuscript. The M.Z.G lab is supported by grants from the European Research Council (669198) and the Welcome Trust (098287/Z/12/Z) and the EU Horizon 2020 Marie Sklodowska-Curie actions (ImageInLife,721537).

## Author contributions

N.C. and A.W. designed and carried out the experiments and data analysis. D.S, K.I.A, and P.T. generated the E-cadherin-GFP transgenic line. N.C, A.W. and M.Z-G. wrote the manuscript. M.Z-G. conceived and supervised the project with the help of N.C.

## Additional information

**Competing interests:** The authors declare no competing interests.

