## [Peer Review File · Nature Communications]

Reviewers' comments:

Reviewer #1 (Remarks to the Author):

This ms addresses the drivers of the morphological changes that the mouse blastocyst undergoes as it transitions into the early egg cylinder stage of development. Using fluorescent tagged marker alleles and live imaging, the authors study the transition of the polar trophectoderm from a squamous epithelium to a columnar shape that also led to a change in the axis of cell division. Together this leads to the beginnings of the inward growth of the extraembryonic ectoderm. They confirm the known role for FGF/ERK signaling in promoting polar TE proliferation and provide evidence that changes in actomyosin contractility are associated with the formation of the polar/mural TE boundary. They also show changes in cell polarity as primitive endoderm cells migrate from the epiblast to overlie the EXE.

Overall the conclusions of the paper are quite reasonable and the use of live imaging is a powerful way to address the underlying morphological events of the peri-implantation period. However, there is a lack of experimental detail about the culture system and the imaging protocols that make it difficult to be sure that the events being recorded are directly comparable to the in vivo events, where the blastocyst is attached to the uterus. It is important that the culture system used provides the best environment for the embryo to develop normally over this critical transition period. The senior author and her group have published improved culture media and approaches to allow the blastocyst to develop over the implantation period (Bedzhov et al 2014 Nature Protocols). It was thus surprising to find that the culture medium used here was not that medium, but a fairly standard tissue culture medium with 20% instead of 10% FBS. It is important for the authors to justify the use of this medium and demonstrate that the embryos do grow, divide, differentiate and show little cell death over the culture period in what would appear to be a less than optimal culture condition.

Additional points to address:

1. Details on the length of time that embryos were imaged in different conditions are hard to extract from the paper- we can only find them from the notes on the images in the figures. It is important to know how long the embryos were imaged, and what was the interval time between images captured. Some of the embryos do not look very happy and it is important to show that this is not a result of phototoxicity.
2. The treatments with inhibitors are also not carefully described. Treatments are listed as 'overnight' which is not a specific interval. How long were the embryos actually in inhibitors and when were they imaged after treatment?

3. The use of the ROCK inhibitor to block actomyosin contractility can be a bit problematic since Rho kinase is involved in several different aspects of cell behavior. Can they confirm with other inhibitors?

4. The section on the role of FGF signaling in promoting polar TE proliferation and Cdx2 expression (l114 on) suggests that this is a novel hypothesis, whereas it has been known for some time that FGF produced first by the ICM and then the epiblast signals to the overlying TE, leading to a very specific domain of phosphor-ERK expression in the overlying TE and EXE (Corson et al 2003). The domain of p-ERK coincides not just with Cdx2 expression but a number of TE-specific transcription factors. A recent paper confirms that loss of FGFR1 leads to altered TE behavior and Cdx2-GFP expression (Kurowski et al 2019, Dev Biol, 446:94). And of course, it is the in vitro replication of this in vivo signaling pathway that allows trophoblast stem cells to self-renew in culture. It would be helpful to incorporate more of this background in this section.

5. The implication in this study is that the inward growth of the EXE is intrinsic to the developing embryo and somehow driven by the directionality of cell divisions as the polar TE cells elongate. However, the extent of the culture period is not really long enough to record much cell division or to see full extension of the egg cylinder. Can the authors really disprove a potential role for the attachment sites of the embryo in the uterus as helping to shape the elongation of the egg cylinder?

6. The section on the spreading of the primitive endoderm (PrE) is not fully elaborated. It focuses on the behavior of the VE but ignores the parietal endoderm, which is actually the first derivative of the PrE to migrate away from the epiblast at the late blastocyst stage. It migrates over the surface of the mural TE and will later form the inner layer of the parietal yolk sac. So it is likely that the PE leads the way and the VE follows to produce a complete layer of cells covering both TE lineages; the EXE and the mural GCs. In fact, the PE migration is apparent in some of the images in the ms, eg Fig 4a, image 2. But in other cases, the mural giant cells and the PE have been damaged and are not included in the images. It would be a much nicer story if the authors could follow all the different transitions that occur as the PrE migrates and develops.

--

Reviewer #2 (Remarks to the Author):

This manuscript describes the movements between the polar and mural trophectoderm (TE), how the TE is regulated by signals from the epiblast (EPI), and last how signals from the primitive endoderm (PE) are involved in directing cell shape changes as it loses exclusive contact with the EPI. This is a really interesting area that has received little attention, and as such, the manuscript would be of significant interest to the readers of Nature Communications. At the same time, the information is so tightly packed that this does not make for an easy read. I suggest the following changes:

Major concerns:

1. Page 3. The manuscript describes a number of striking correlations, but sometimes the results are just that, correlations. How can the authors be certain that it is the CDX2 gradient that causes the actomyosin boundary? Could the authors mosaically mis-express CDX2 and examine the effect on the actomyosin network?
2. Page 4. The authors implicate FGF as a signaling pathway that may regulate TE development. However, they should be more precise as to which ligands and receptors are involved, especially as a number of recent reports (not cited) focus on this, some of which implicate FGF signaling in TE development. For instance, in the blastocyst, FGF4 is the only EPI ligand, but more ligands are expressed in the EPI following implantation. FGFR1 is expressed in all lineages and FGFR2 is expressed in both the TE and PRE. These receptors are also expressed in both the mural and polar TE, and thus could receive FGF4 signals from the EPI. FGFR3 and FGFR4 come up somewhat later, at least in the PrE. What happens if the authors culture embryos in the presence of higher amounts of FGF?
3. pERK, while an indicative readout of FGF signaling, cannot be viewed as a definitive one, as FGF signaling induces more than just ERK1/2 activation, and ERK1/2 can also be activated by more pathways than just FGFRs. Have the authors analyzed for instance PI3K signaling?

Minor concerns:

1. I can understand how a tight actomyosin can indicate a cell boundary. However, prevention of cell mixing can also be regulated by other processes such as repulsive cues. The authors should at least mention such possibilities.
2. I am wondering if the initial data on using the ROCK inhibitor would not be better if moved up in the text, when discussing actomyosin boundaries.

Reviewers' comments:

Reviewer #1 (Remarks to the Author):

This ms addresses the drivers of the morphological changes that the mouse blastocyst undergoes as it transitions into the early egg cylinder stage of development. Using fluorescent tagged marker alleles and live imaging, the authors study the transition of the polar trophectoderm from a squamous epithelium to a columnar shape that also led to a change in the axis of cell division. Together this leads to the beginnings of the inward growth of the extraembryonic ectoderm. They confirm the known role for FGF/ERK signaling in promoting polar TE proliferation and provide evidence that changes in actomyosin contractility are associated with the formation of the polar/mural TE boundary. They also show changes in cell polarity as primitive endoderm cells migrate from the epiblast to overlie the EXE.

We would like to thank the reviewer for these comments and all the feedback below, which has helped us to improve our manuscript.

Here, we wish to make just one point - that we do not describe "inward growth of the extra-embryonic ectoderm". Rather, we show that the expansion of the extra-embryonic ectoderm (EXE) is driven by cell shape changes and polarised cell divisions within the polar TE instead. These events are followed by apical constriction mediated folding of the tissue during extra-embryonic cavity formation. Thus, the inward movement of the tissue is driven by tissue folding and not growth.

We also do not report "migration of primitive endoderm cells to overlie the ExE". In contrast, we show that this process does not require cell migration, as we show through inhibition of Rac (Fig. S11). Furthermore, we demonstrate that primitive endoderm (PE) cells are enclosed within the Reichert's membrane (Fig. 5e), which does not support an active migration mechanism for PE spreading during visceral endoderm formation.

Our findings indicate that the primitive endoderm spreads to cover the epiblast (EPI) and the ExE during the formation of the visceral endoderm in response to forces

generated by the expansion of polar TE and subsequent ExE folding, which results in the repositioning of the embryo towards the abembryonic pole (Fig. 6, Fig. S9).

To make these points clearer, we now add some detailed discussion of this process in the discussion section.

Overall the conclusions of the paper are quite reasonable, and the use of live imaging is a powerful way to address the underlying morphological events of the peri-implantation period.

We thank the reviewer for this comment.

However, there is a lack of experimental detail about the culture system and the imaging protocols that make it difficult to be sure that the events being recorded are directly comparable to the in vivo events, where the blastocyst is attached to the uterus. It is important that the culture system used provides the best environment for the embryo to develop normally over this critical transition period.

We agree with the reviewer's statement that the culture system used has to provide the best environment possible for the embryo to develop normally over this critical transition period. This is why we have confirmed all our live imaging data by also analysing embryos fixed directly upon recovery from the mother, as we indicate in the manuscript.

We now also include a more detailed description of the culture system and imaging protocols. In addition, we now provide a detailed protocol for live imaging as a supplementary methods file.

The senior author and her group have published improved culture media and approaches to allow the blastocyst to develop over the implantation period (Bedzhov et al 2014 Nature Protocols). It was thus surprising to find that the culture medium used here was not that medium, but a fairly standard tissue culture medium with 20% instead of 10% FBS. It is important for the authors to justify the use of this medium and demonstrate that the embryos do grow, divide, differentiate and show

little cell death over the culture period in what would appear to be a less than optimal culture condition.

Here, we would like to point out that the referee makes a mistake. Our culture medium is supplemented with 20% serum as it has been previously reported by our group (Bedzhov et al., 2014 Nature Protocols). In Bedzhov et al. 2014 the culture medium is supplemented with 20% FBS and not 10% as suggested by the reviewer. The only difference in the medium used in this study from the Bedzhov study is that the medium is not supplemented with hormones because our embryos are already implanting.

Below are some examples of implanted E4.8 embryos cultured for 20h (Red:DAPI,Cyan:F-actin) showing normal development and embryogenesis.

Additional points to address:

1. Details on the length of time that embryos were imaged in different conditions are hard to extract from the paper- we can only find them from the notes on the images in the figures. It is important to know how long the embryos were imaged, and what was the interval time between images captured. Some of the embryos do not look very happy and it is important to show that this is not a result of phototoxicity.

We have added this information to the text as requested by the reviewer. We also include the time interval and number of time points in each legend for the supplementary movies.

All the events that we observed during live imaging were confirmed by analysis of embryos fixed immediately upon recovery from the decidua. Thus, what we have observed under the microscope is not an artefact of the embryo culture and imaging conditions.

In support of this, we have previously shown that EXE folding during cavity formation is evident in embryos developing in vitro and in vivo (Christodoulou et al., 2018). This is now described in detail in supplementary methods. We do not observe phototoxicity or cell death and embryos develop normally under our conditions (Movies 1-10).

2. The treatments with inhibitors are also not carefully described. Treatments are listed as ‘overnight’ which is not a specific interval. How long were the embryos actually in inhibitors and when were they imaged after treatment?

We have provided this information in the Methods section. In Figure 4c, we have described the experimental protocol for ex-vivo inhibitor treatment. The same applies to Figure 6h.

3. The use of the ROCK inhibitor to block actomyosin contractility can be a bit problematic since Rho kinase is involved in several different aspects of cell behavior. Can they confirm with other inhibitors?

We thank the referee for this insight. As a second approach to inhibit actomyosin contractility, we have now used blebbistatin(-), a specific myosin II ATPase inhibitor, which inhibits actomyosin contractility. When we cultured E4.5 blastocysts for 20h in the presence of 50um blebbistatin (-), we found that formation of the polar/mural TE tissue boundary becomes defective and polar TE expansion is inhibited. Thus, blebbistatin treatment phenocopies the phenotype we obtained upon ROCK inhibition. We present these new data in Fig. S6b-d.

In addition, we show that ROCK inhibition results in reduction of myosin phosphorylation at the interface between polar and mural TE (Fig. S6a)

4. The section on the role of FGF signaling in promoting polar TE proliferation and Cdx2 expression (l114 on) suggests that this is a novel hypothesis, whereas it has been known for some time that FGF produced first by the ICM and then the epiblast signals to the overlying TE, leading to a very specific domain of phosphor-ERK expression in the overlying TE and EXE (Corson et al 2003). The domain of p-ERK coincides not just with Cdx2 expression but a number of TE-specific transcription factors. A recent paper confirms that loss of FGFR1 leads to altered TE behavior and Cdx2-GFP expression (Kurowski et al 2019, Dev Biol, 446:94). And of course, it is the in vitro replication of this in vivo signaling pathway that allows trophoblast stem cells to self-renew in culture. It would be helpful to incorporate more of this background in this section.

To respond to the reviewer's suggestion, we now expanded the discussion of FGF signalling in the revised version of the manuscript.

Regarding the referee's comments on Kurowski et al 2019, Dev Biol, 446:94. This paper, which we like very much, was not published when our work was submitted. We wish to say that it has been well known that FGF signalling is necessary for primitive endoderm specification from E3.5-E4.5. Fgfr1 ^{-/-} embryos do not specify PE, thus the phenotype observed in the terminal differentiation of mural TE might be due to the absence of primitive endoderm. Furthermore, Fgfr1^{-/-} blastocysts have a smaller cavity and this might affect the generation of the FGF signalling gradient within the TE. Therefore, the phenotype observed in Fgfr1^{-/-} embryos in the mural TE cannot be attributed solely to TE behaviour as the PE is not specified. Kurowski et al 2019 did not explore these possibilities.

5. The implication in this study is that the inward growth of the EXE is intrinsic to the developing embryo and somehow driven by the directionality of cell divisions as the polar TE cells elongate. However, the extent of the culture period is not really long enough to record much cell division or to see full extension of the egg cylinder

We want to apologise if we did not make it clear in the previous version of the manuscript that we observe an inward movement of the ExE due to tissue folding driven by apical constriction, and therefore not as a result of “the directionality of cell divisions”. We have now added new data (Fig. S9) to further demonstrate this and included a paragraph in the discussion section about this.

Briefly, the ExE folding process occurs in a window of 2-3 hours (Fig. 6 a,b) and is driven by apical constriction and not cell divisions. The polarised cell divisions driving polar TE expansion occur before the folding process takes place (Figure 3d, 4i, 5c, 6j).

We would like to note that our study focuses on the formation of the egg cylinder, which takes place at E5.0, and not its elongation, which occurs between E5.25-E6.0.

Can the authors really disprove a potential role for the attachment sites of the embryo in the uterus as helping to shape the elongation of the egg cylinder?

When we initiated this study, we had the same concern. However, from our in vitro culture experiments it is clear that embryos can form egg cylinders in the absence of maternal tissues (Fig. 6a,h, Fig. S10c-d, Fig. S11). This is in agreement with a previous study from our lab showing that embryos cultured in vitro within hanging drops (thus in the absence of external mechanical cues) can form egg cylinders and specify an anteroposterior axis (Bedzhov et al., 2015).

6. The section on the spreading of the primitive endoderm (PrE) is not fully elaborated. It focuses on the behavior of the VE but ignores the parietal endoderm, which is actually the first derivative of the PrE to migrate away from the epiblast at the late blastocyst stage. It migrates over the surface of the mural TE and will later form the inner layer of the parietal yolk sac. So it is likely that the Parietal Endoderm leads the way and the VE follows to produce a complete layer of cells covering both TE lineages; the EXE and the mural GCs. In fact, the Parietal Endoderm migration is apparent in some of the images in the ms, eg Fig 4a, image 2. But in other cases, the mural giant cells and the PE have been damaged and are not included in the images. It would be a much nicer story if the authors could follow all the different transitions that occur as the PrE migrates and develops.

We thank the reviewer for this comment. This made us consider the contribution of parietal endoderm migration in the blastocyst to egg cylinder transition. We now provide data showing that parietal endoderm migration precedes the cell shape changes within the PE and the polar TE, which result in the expansion of the polar trophoctoderm and the spreading of the primitive endoderm. (Fig. 5d). Thus, parietal endoderm migration can not contribute to egg cylinder formation as it takes place before the first morphogenetic event of blastocyst to egg cylinder formation.

Additionally, in Dystroglycan 1 knock-out mice the Reichert's membrane (the substrate for parietal endoderm migration and a mechanical scaffold) does not form and parietal endoderm does not migrate. Despite these defects Dag1 ^{-/-} embryos form egg cylinders (Williamson et al., 1997). This suggests that parietal endoderm migration is not necessary for egg cylinder formation.

Reviewer #2 (Remarks to the Author):

This manuscript describes the movements between the polar and mural trophoctoderm (TE), how the TE is regulated by signals from the epiblast (EPI), and last how signals from the primitive endoderm (PE) are involved in directing cell shape changes as it loses exclusive contact with the EPI. This is a really interesting area that has received little attention, and as such, the manuscript would be of significant interest to the readers of Nature Communications. At the same time, the information is so tightly packed that this does not make for an easy read. I suggest the following changes:

We would like to thank the reviewer for these supportive comments.

Major concerns:

1. Page 3. The manuscript describes a number of striking correlations, but sometimes the results are just that, correlations. How can the authors be certain that it is the CDX2 gradient that causes the actomyosin boundary?

We thank the referee for this comment and we have revised the manuscript accordingly. We interpret our data to mean that differential trophoctoderm (TE) fate

acquisition (multipotent CDX2+ vs differentiated CDX2- cells) is responsible for boundary formation within this tissue. We use CDX2 as a marker of the TE's multipotent character, but we cannot be sure whether CDX2 regulates tissue boundary formation directly.

Specifically, we propose that differential fate acquisition in polar and mural TE leads to the formation of a tissue boundary based on the findings that:

- 1) The mural/polar TE boundary is formed only upon the appearance of differential fate in the TE.
- 2) Inhibition of FGF signalling (upon treatment of embryos with FGFR inhibitor) results in defective TE tissue boundary formation - all TE cells have the same fate as they are now all differentiated (Fig. 4d-h).

We now make this point more clearly as we have space for a Discussion section.

Could the authors mosaically mis-express CDX2 and examine the effect on the actomyosin network?

We expect that CDX2 alone will not be sufficient to maintain multipotency within the mural TE since induced trophoblast stem cells (after overexpression of CDX2 in embryonic stem cells) can be maintained in culture in an undifferentiated state only in the presence of FGF and MEFs or MEF conditioned medium (Blij et al., 2015; Niwa et al., 2005). Technically it will be very difficult to locally express CDX2 and activate FGF signalling in the mural TE.

2. Page 4. The authors implicate FGF as a signaling pathway that may regulate TE development. However, they should be more precise as to which ligands and receptors are involved, especially as a number of recent reports (not cited) focus on this, some of which implicate FGF signaling in TE development. For instance, in the blastocyst, FGF4 is the only EPI ligand, but more ligands are expressed in the EPI following implantation. FGFR1 is expressed in all lineages and FGFR2 is expressed in both the TE and PRE. These receptors are also expressed in both the mural and poplar TE, and thus could receive FGF4 signals from the EPI. FGFR3 and FGFR4 come up somewhat later, at least in the PrE.

We thank the reviewer for this comment and have now expanded discussion of FGF signalling pathways in the paper (as only now do we have a space for a Discussion section). In addition, in response to the reviewer's suggestion, we have analysed sequencing data of epiblast (EPI) at the time of implantation using the dataset we generated previously (Shahbazi et al., 2017). This has revealed two FGF ligands produced by the EPI at the time of implantation: FGF4 and FGF5 (Fig. S4a).

What happens if the authors culture embryos in the presence of higher amounts of FGF?

We thank the reviewer for this suggestion. In response to this point, we performed the following experiment.

We recovered E3.5 blastocysts, cultured them for 24h in KSOM, and then transferred the blastocysts into IVC medium in the presence or absence of 1000ng/ml FGF with 1um Heparin. Blastocysts were cultured for an additional 24h and then fixed. This experimental set up was used to avoid the impact of FGF treatment on primitive endoderm (PE) specification.

Our results show that FGF treatment results in defective differentiation of mural TE cells as indicated by the absence of a CDX2 gradient in the TE of treated blastocysts (Fig. S5). This data (together with our experiments using FGFR inhibition) indicates that FGF signalling provided by the EPI results in differential fate acquisition between polar and mural TE.

3. pERK, while an indicative readout of FGF signaling, cannot be viewed as a definitive one, as FGF signaling induces more than just ERK1/2 activation, and ERK1/2 can also be activated by more pathways than just FGFRs. Have the authors analyzed for instance PI3K signaling?

We thank the reviewer for this insight. Based on the reviewer's comment we carried out additional experiments and now show that pAKT (active PI3K signalling) is present only in polar TE, similar to pERK. This confirms that FGF signalling is specifically activated in the polar TE.

Minor concerns:

1. I can understand how a tight actomyosin can indicate a cell boundary. However, prevention of cell mixing can also be regulated by other processes such as repulsive cues. The authors should at least mention such possibilities.

As requested by the reviewer, we now discuss the different possibilities for generation of localised actomyosin contractility during tissue boundary formation (Discussion, page 9)

2. I am wondering if the initial data on using the ROCK inhibitor would not be better if moved up in the text, when discussing actomyosin boundaries.

We thank the reviewer for this suggestion. We thought about it but in the end decided to have the functional experiment with the ROCK inhibitor in the same section as FGF inhibition in order to highlight the role of TE tissue boundary in subsequent embryo development. We hope however that our revisions have made the whole manuscript much clearer.

References

- Bedzhov, I., Bialecka, M., Zielinska, A., Kosalka, J., Antonica, F., Thompson, A.J., Franze, K., and Zernicka-Goetz, M. (2015). Development of the anterior-posterior axis is a self-organizing process in the absence of maternal cues in the mouse embryo. *Cell Res* 25, 1368-1371.
- Blij, S., Parenti, A., Tabatabai-Yazdi, N., and Ralston, A. (2015). Cdx2 efficiently induces trophoblast stem-like cells in naive, but not primed, pluripotent stem cells. *Stem Cells Dev* 24, 1352-1365.
- Christodoulou, N., Kyprianou, C., Weberling, A., Wang, R., Cui, G., Peng, G., Jing, N., and Zernicka-Goetz, M. (2018). Sequential formation and resolution of multiple rosettes drive embryo remodelling after implantation. *Nat Cell Biol*.
- Niwa, H., Toyooka, Y., Shimosato, D., Strumpf, D., Takahashi, K., Yagi, R., and Rossant, J. (2005). Interaction between Oct3/4 and Cdx2 determines trophectoderm differentiation. *Cell* 123, 917-929.
- Shahbazi, M.N., Scialdone, A., Skorupska, N., Weberling, A., Recher, G., Zhu, M., Jedrusik, A., Devito, L.G., Noli, L., Macaulay, I.C., *et al.* (2017). Pluripotent state transitions coordinate morphogenesis in mouse and human embryos. *Nature* 552, 239-243.
- Williamson, R.A., Henry, M.D., Daniels, K.J., Hrstka, R.F., Lee, J.C., Sunada, Y., Ibraghimov-Beskrovnaya, O., and Campbell, K.P. (1997). Dystroglycan is essential for early embryonic

development: disruption of Reichert's membrane in Dag1-null mice. Hum Mol Genet 6, 831-841.

REVIEWERS' COMMENTS:

Reviewer #1 (Remarks to the Author):

The authors have carefully addressed the specific comments of the referees and added some experimental details and data to support their findings. I would just like to take some issue with their overinterpretation of my comments on 'growth' of the trophoblast and 'migration' of the PE. I did understand that they were largely showing reorganization and folding of the trophoblast resulting in inward growth into the blastocoelic cavity. Ditto - migration is what is observed- whether it is active or not, needs to be determined.

I also take issue with their claim that we did not know that FGF played an important role in postimplantation TE development- they are really ignoring a lot of different data to come to this definitive claim (I85). And it really is not necessary to be so dogmatic- this current study does indeed add some new insights into how the FGF signal from the epiblast is received and interpreted by the Trophoblast to drive egg cylinder development

--

Reviewer #2 (Remarks to the Author):

The authors have addressed my comments and the manuscript is now much improved. I think this will make a valuable contribution to the field and will be read with interest.

I still think some additional references should have been cited, but this is not critical.

REVIEWERS' COMMENTS:

Reviewer #1 (Remarks to the Author):

The authors have carefully addressed the specific comments of the referees and added some experimental details and data to support their findings.

We thank the reviewer for the overall feedback.

I would just like to take some issue with their overinterpretation of my comments on 'growth' of the trophoblast and 'migration' of the PE. I did understand that they were largely showing reorganization and folding of the trophoblast resulting in inward growth into the blastocoelic cavity. Ditto - migration is what is observed- whether it is active or not, needs to be determined.

The Rac-inhibitor experiments (Supplementary figure 11) suggest that active cell migration is not involved in the blastocyst to egg cylinder transition. Therefore, we propose that this morphogenetic process does not involve active cell migration and that the observed cell movement is a result of a responses to the forces generated during trophectoderm expansion and folding (details in the Discussion section).

I also take issue with their claim that we did not know that FGF played an important role in postimplantation TE development- they are really ignoring a lot of different data to come to this definitive claim (185). And it really is not necessary to be so dogmatic- this current study does indeed add some new insights into how the FGF signal from the epiblast is received and interpreted by the Trophoblast to drive egg cylinder development

This sentence has been removed from the revised version of the manuscript.

--

Reviewer #2 (Remarks to the Author):

The authors have addressed my comments and the manuscript is now much improved. I think this will make a valuable contribution to the field and will be read with interest.

I still think some additional references should have been cited, but this is not critical.

We thank the reviewer for the overall feedback. We have now added additional references and discuss in more detail the role of FGF signalling during mouse pre-implantation development in the revised manuscript.